# LocalGAN: Modeling Local Distributions for Adversarial Response Generation

## Abstract

This paper presents a new methodology for modeling the local semantic distribution of responses to a given query in the human-conversation corpus, and on this basis, explores a specified adversarial learning mechanism for training Neural Response Generation (NRG) models to build conversational agents. The proposed mechanism aims to address the training instability problem and improve the quality of generated results of Generative Adversarial Nets (GAN) in their utilizations in the response generation scenario. Our investigation begins with the thorough discussions upon the objective function brought by general GAN architectures to NRG models, and the training instability problem is proved to be ascribed to the special local distributions of conversational corpora. Consequently, an energy function is employed to estimate the status of a local area restricted by the query and its responses in the semantic space, and the mathematical approximation of this energy-based distribution is finally found. Building on this foundation, a local distribution oriented objective is proposed and combined with the original objective, working as a hybrid loss for the adversarial training of response generation models, named as LocalGAN. Our experimental results demonstrate that the reasonable local distribution modeling of the query-response corpus is of great importance to adversarial NRG, and our proposed LocalGAN is promising for improving both the training stability and the quality of generated results.

## 1 Introduction

End-to-End generative conversational agents (a.k.a., generative Chat-bots) are believed to be practicable on the basis of the Sequence-to-Sequence (Seq2Seq) architecture (Sutskever et al., 2014) trained with large amounts of human-generated conversation sessions (Shang et al., 2015a; Sordoni et al., 2015), and this task is named as Neural Response Generation (NRG). Similar to the Neural Machine Translation (NMT) approaches (Bahdanau et al., 2014; Wu et al., 2016), the deep Seq2Seq models are expected to directly generate appropriate and meaningful responses according to the input query. Compared to the success of NMT systems, the application progress of NRG models is not satisfying at present due to the "safe response" problem (Li et al., 2016). That is, most of the generated responses are boring and meaningless, which blocks the continuation of conversations. Indeed, eliminating "safe responses" is the essential task of NRG models. Thus, various methods have been considered to address this problem (Li et al., 2016; Xu et al., 2017; Li et al., 2017; Xing et al., 2017; Pandey et al., 2018; Zhang et al., 2018a; Du et al., 2018).

More recently, Generative Adversarial Nets (GAN) (Goodfellow et al., 2014) have been introduced to eliminate "safe responses" (Li et al., 2017; Xu et al., 2017). Basically, this methodology is reasonable since the GAN framework involves an adversarial discriminator that helps NRG models leap out of the shortsighted state of minimizing the empirical risk on word distribution, by providing feedback on real samples from the model generated ones. Despite the improvement on the diversity, the adversarial training process of GAN based response generation models is generally unstable and sensitive to the training strategy (Yu et al., 2017).

The unstable convergence problem is largely ascribed to the complicated data distribution in practical scenarios (Arora et al., 2017; Arora & Zhang, 2017). For the response generation oriented GAN models, in particular, the data distribution appears to be much more complicated. Fundamentally, an essential characteristic of conversation data is that, for each given query, there always exists a

group of semantically-diverse responses, rather than the semantically-unified ones. Furthermore, the response groups of two different queries tend to keep great divergences in the semantic space. In this scenario, the discriminator needs to consider the distributions of the generated result in the semantic space, rather than simply examining whether one single sample comes from the generator or the original dataset, so as to make the generator sense the distribution of conversation dataset in the adversarial training procedure.

This paper aims at presenting a specific adversarial training schema for neural response generation. Beginning with the investigation on the reason for the unsatisfying performance of GANs on the NRG task, we find the upper-bound of the current GAN learning strategies taking query-response pairs as the independent training samples. On this basis, we claim that the training schema, including the actual adversarial strategy and the overall loss function, should be re-defined to agree with the distribution of NRG training samples in the semantic space, rather than roughly adopting the GAN framework designed for generating images.

Consequently, we describe the distributional state of the given query and the corresponding responses with the free energy defined on the basis of Deep Boltzmann Machines (DBM) (Salakhutdinov & Hinton, 2009). In this way, we can quantify the formation process of generating the response set with the topic restriction of the given query. From the perspective of free energy, this paper proposes a new cost function to measure the expansion degree of the responses in the local area of the real-valued semantic space. Cooperating with the traditional implicit density discriminating loss of GAN, the proposed cost actually provides an explicit density approximation for the local distribution of each response cluster. Thus, the adversarial learning procedure can be expected to be more stable with better response generation results obtained[1].

## 2 THE LIMITATION OF GENERAL GAN IN THE NRG SCENARIO

According to (Goodfellow et al., 2014), the standard GAN framework contains a generator $G$ and a discriminator $D$, which are trained by an iterative adversarial learning procedure based on the following objective function:

$$J^{(D)} = \mathbb{E}_{x \sim p_d} \left[ \log D(x) \right] + \mathbb{E}_{z \sim p_z} \left[ \log(1 - D(G(z))) \right] \tag{1}$$

$$J^{(G)} = \mathbb{E}_{z \sim p_z} \left[ \log D(G(z)) \right] \tag{2}$$

where $p_z$ denotes the prior on input noise variables, and $p_d$ is the true data distribution.

It should be noted that the GAN tries to learn the manifold of a given dataset (Khayatkhoei et al., 2018; Kumar et al., 2017), and the discriminator $D$ actually provides a metric for judging whether the results generated according to $z$ fits the expected manifold or not. In the NRG scenario, a naive Seq2Seq model without the guidance signal from $D$ can not capture the data manifold of the real query-response corpus, which is one of the major facts the safe-response problem can be ascribed to. Assuming that there exists an oracle discriminator with the ability of distinguishing the generated fake samples from the ground-truth ones, by mapping each query-response pair $(q, r)$[2] to a confidence score $s$, and meanwhile, it can be assume that any practically existing discriminator of GAN gives the confidence $\tilde{s}$ to $(q, r)$. If the practical discriminator can make $\tilde{s} \rightarrow s$, the generator will obtain more meaningful guidance for the better generation. That is, to improve the capability of GAN-NRG, it is wise to construct more powerful discriminators for more reasonable $J^{(G)}$.

Now let's pay attention to the actual change of NRG models brought by GAN. In the generative conversation agent scenario, $G(z)$ is corresponding to a generated response $\tilde{r}$ to a given query $q$. Thus, the objective of the generator in the GAN based NRG model can be simply formulated as:

$$J^{(G)} = \mathbb{E}_{(q, \tilde{r}) \sim p_g} \left[ \log D(q, \tilde{r}) \right] \tag{3}$$

The training of the generator is actually the procedure to maximize $J^{(G)}$ toward $\mathbb{E}_{(q, r) \sim p_d} \left[ \log D(q, r) \right]$, so as to generate realistic responses according to given queries. Thus, in

---

[1] The code of our proposed model LocalGAN can be found in `https://github.com/Kramgasse49/local_gan_4generation`

[2] Here $q$ and $r$ represent the vectorized query and its response. We take the simple embedding-averaging based method to transform texts into vectors. Besides, due to our adopted text vectorizing method, the pre-training phase of the Deep Boltzmann Machine takes some specified trick detailed in Appendix A.

the context of adversarial learning, $J^{(G)}$ should satisfy the following inequality:

$$J^{(G)} \leqslant \mathbb{E}_{(q,r)\sim p_d} [\log D(q,r)] \tag{4}$$

However, it is well known that a conversational dataset should not be simply taken as a collection $\{(q,r)\}$ composed of independent query-response pairs. Instead, to each given query $q$, there exists a finite set of corresponding responses $R_q = \{r_i\}$. In this case, it is of great necessity to consider the whole training dataset as a collection of $R_q$, which takes the form of a number of clusters with their own local distributions in the semantic space. And we can rewrite the joint distribution $p_d$ in (4) as $p(q)p(r|q)$ and assume every corresponding response to a query follows equal-probability distribution, which means that $p(r|q) = \frac{1}{|R_q|}$. Thus, in the real NRG scenario, on the basis of Equation 3 and Inequation 4, $J^{(G)}$ follows the inequality as below:

$$J^{(G)} \leqslant \sum_q \mathbb{E}_{(q,r)\sim q,R_q} [\log D(q,r)]$$

$$= \sum_q \sum_{r\in R_q} p(q)\frac{1}{|R_q|} [\log D(q,r)] \leqslant \sum_q p(q) \log \left[ \frac{1}{|R_q|} \sum_{r\in R_q} D(q,r) \right] \tag{5}$$

where $p(q)$ denotes the probability of the query $q$, and $R_q$ is defined above.

According to Equation 5, the upper bound of $J^{(G)}$ is obtained, in which the $\log \left[ \frac{1}{|R_q|} \sum_{r\in R_q} D(q,r) \right]$ part is the essence. Apparently, the expression $\frac{1}{|R_q|} \sum_{r\in R_q} D(q,r)$ indicates the mean value of the confidence scores given by the discriminator to each member of the response set $R_q$ to a given query $q$. Moreover, it should be noted that current studies tend to utilize semantic relevance oriented models to build the discriminators of GAN (Xu et al., 2017; Li et al., 2017). Consequently, $D(q,r)$ can be actually considered as the spatial relationship of $q$ and $r$ in the semantic space. Thus, $\frac{1}{|R_q|} \sum_{r\in R_q} D(q,r)$ stands for the spatial center of all the responses within $R_q$. That is, the optimization process of adversarial learning upon conversational datasets will make the generated responses approach to the center of each local distribution of $R_q$ corresponding to each given dependent query.

The practical value of this change lies in that, intuitively, the GAN architecture forces the generation to pay attention to the local distributions of the individual response clusters, rather than taking the $(q,r)$-pairs as an entirety. According to the thorough studies on the safe responses of NRG models (Li et al., 2016; Xu et al., 2017; Zhang et al., 2018a; Pandey et al., 2018), it can be inferred that the general Seq2Seq will fall into the divergence state of generating the patterns with the maximum probabilities taking account of the entire dataset, ignoring the individual-difference of each query. By introducing the implicit loss focusing on the response clusters, GAN makes the divergence of the generator much closer to the 'local patterns' rather than the general patterns, and thus the higher diversity can be expected and observed (Xu et al., 2017; Li et al., 2017).

The problem turns to: Is the upper bound in Equation 5 powerful enough? Apparently, there exists an obvious gap between the 'local patterns' and the vivid and interesting generated responses. The upper bound only focuses on the mean of each response set, but the local distribution (or the actual 'shape') of each cluster has not been taken into account. This situation does not change when the cost function of adversarial training is defined by Wasserstein GANs (WGAN) (Arjovsky et al., 2017) with 1-Lipschitz function $f$:

$$J_W^{(G)} = \mathbb{E}_{(q,\tilde{r})\sim p_g} [f(q,\tilde{r})] \tag{6}$$

Intuitively, it is of paramount importance to estimate both the 'location' and the 'shape' of the response set $R_q$ in the semantic space (indeed, $\frac{1}{|R_q|} \sum_{q,r} D(q,r)$ is only relative with 'location'), so as to determine the optimization objective of adversarial training. Consequently, we have two critical problems to be discussed and addressed in the following sections:

- How to describe the state of the response set $R_q$ with a given query $q$ in the semantic space?
- Taking account of the reasonable state modeling of $(q, R_q)$, what is the loss function for adversarial training to generate responses?

## 3 MODELING THE STATE OF THE LOCAL DISTRIBUTION FOR RESPONSES

As mentioned above, the semantic one-to-many relationship between queries and responses makes it necessary to model the local distribution of the response cluster $R_q$ corresponding to each query in the space, and it is paramount to turn to the fitting of each local distribution in the adversarial learning procedure, rather than considering each $(q, r)$-pair as an independent sample. Basically, this issue equals to the task of reasonably modeling the state of $(q, R_q)$ in the semantic space, by considering each $(q, R_q)$ as a systematic entirety and assigning the state of the entirety with probabilistic distribution. The additional major challenge of this task is, indeed, we have to infer the state of a local area in the semantic space from a group of finite samples, since it is impossible to sample all the possible responses to a query from the given corpus, regardless of the corpus size.

### 3.1 REPRESENTING LOCAL DISTRIBUTIONS WITH QUERY-RESPONSE ORIENTED FREE ENERGY

In this part, we typically take an energy based statistical model, the Average Free Energy (Hinton & Zemel, 1994; Friston et al., 2006; Ngiam et al., 2011; Friston et al., 2012), to describe the state of the local distribution of $(q, R_q)$ in the semantic space, for the reasons that: a) energy based models are considered as a promising avenue towards learning explicit generative models (LeCun et al., 2006; Le Roux & Bengio, 2008), by representing data distributions without any prior assumptions; and b) energy based models can be trained in the unsupervised way, and the energy functions of such models have the potential to estimate the state of generative models (Zhao et al., 2016).

At first, the free energy of a given query-response pair $(q, r)$ can be defined as following:

$$F(q, r) = -\log \left[ \sum_H \exp(E(q, r, H)) \right] \tag{7}$$

where $E(q, r, H)$ stands for the energy function defined according to the relationship of the query $q$ and its response $r$ via the hidden variable $H$.

We employ the Deep Boltzmann Machine (DBM) (Salakhutdinov & Larochelle, 2010; Smolensky, 1986; Hinton & Salakhutdinov, 2006) to implement $E(q, r, H)$, as illustrated by Figure 1. The reason for this choice lies in that, from the view of conversational agents, the meaningful query and the corresponding response are generally considered to maintain strong semantic relevance. Thus, the query and response can be mutually transformed into each other, which is supported by the considerable amount of studies on response generation (Shang et al., 2015b; Shao et al., 2017; Zhang et al., 2018a; Baheti et al., 2018) and question generation (Du et al., 2017; Zhao et al., 2018; Sun et al., 2018). Without loss of generality, the pairwise semantic relationship of the query $q$ and the corresponding response $r$ can be

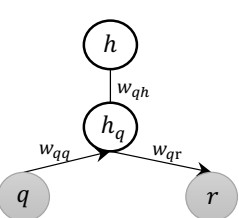

Figure 1: The DBM for modeling the semantic relationship of Query-Response pairs.

modeled by a two-layer DBM. The bottom layer is actually an abstract version of Seq2Seq models, in which a response $r$ can be generated based on a hidden variable $h_q$, and $h_q$ depends on the given query $q$ theoretically. In the top layer, $h_q$ conditionally depends on a hyper hidden variable $h$. Figure 1 illustrates the Deep Boltzmann Machine for modeling the semantic relationship of a query and its responses.

Following (Salakhutdinov & Hinton, 2009), on the basis of the DBM in Figure 1, the energy of the state $\{(q, r), H\}$ is defined as:

$$E(q, r, H) = E(q, r, h_q, h) = -h_q^T W_{qr} r - h_q^T W_{qq} q - h^T W_{qh} h_q \tag{8}$$

where $q$ denotes the query, $r$ stands for the response and $H = \{h_q, h\}$ represents the hidden units. $W_{qr}$, $W_{qq}$ and $W_{qh}$ stand for the weights on the corresponding connections of the query, response and the hidden variables respectively in the graph model shown by Figure 1.

Consequently, we can define the average free energy of the query $q$ and its response set $R_q$ as follows:

$$F(q, R_q) = \frac{1}{|R_q|} \sum_{r_i \in R_q} F(q, r_i) \tag{9}$$

For better conducting the following discussion, we further define the energy difference between response $r_i$ and $r_j$ as:

**Definition 3.1** (**Scaled Energy Difference**).

$$\Delta_{q, r_i, r_j} = \frac{F(q, r_i) - F(q, r_j)}{F(q, R_q)} \tag{10}$$

Meanwhile, it is necessary to assign a spatial intuition to $R_q$ in the semantic space by defining:

**Definition 3.2** (**Response Cluster**). In the semantic space, the meaningful responses to the given query $q$ lie in a restricted region (e.g., a hyper sphere), which can be named as the **Response Cluster**, in which $F(q, r)$ can be taken as the distance from a response $r$ to the cluster center.

## 3.2 ESTIMATION OF $F(q, R_q)$

Basically, the DBM in Figure 1 provides the definition of the energy function $E(q, r, H)$ of the free energy given in Equation 8. Thus, the local distribution state of the responses $R_q$ with the given query $q$ can be mathematically described by $F(q, R_q)$ based on Equation 7 - 9. In practice, however, this procedure is not operable yet because the computation of $F(q, R_q)$ requires all the response in $R_q$, and it is impractical to perform the exhaustive enumeration over all the possible responses of the given query, regardless of the amount of the training query-response pairs. Consequently, it is highly necessary to approximate $F(q, R_q)$ under some reasonable assumptions.

Considering the response cluster defined in Definition 3.2, we can naturally assume that the response random variable $r$ to a given query $q$ follows multivariate normal distribution with mean $r_c$ and covariance matrix $\Sigma$, where $r_c$ and $\Sigma$ is only determined by query $q$. Afterwards, the observed $R_q$ can be considered as a realization of the $|R_q|$-variate random variable $(r_1, \cdots, r_{|R_q|})$ for $r_i, 1 \leq i \leq |R_q|$, i.i.d random variables drawn from the distribution $N(r_c, \Sigma)$. Consequently, an executable approximation of $F(q, R_q)$ can be obtained based on the following Lemmas and Theorem[3].

**Lemma 1.** *If $\mathbb{E}[F(q, r)] < \infty$, $F(q, R_q)$ converges almost surely to the expected value $\mathbb{E}[F(q, r)]$, as $|R_q|$ goes to infinity. That is,*

$$P\left( \lim_{|R_q| \to \infty} F(q, R_q) = \mathbb{E}[F(q, r)] \right) = 1. \tag{11}$$

Lemma 1 indicates that for any $\epsilon > 0$ there exists $N(\epsilon)$ such that if $|R_q| > N(\epsilon)$ then the inequality $|F(q, R_q) - \mathbb{E}[F(q, r)]| < \epsilon$ holds. In other words, $F(q, R_q)$ can be very close to $\mathbb{E}[F(q, r)]$ if the responses are sufficient.

**Lemma 2.**

$$\lim_{\mathbb{E}\|r - r_c\|_2 \to 0} |\mathbb{E}[f(r)] - f(r_c)| = 0 \tag{12}$$

Practically, the expected Euclidean distance between random variable $r$ and $r_c$ can not be zero. Thus, the fact conveyed by Lemma 2 is that, actually, if the expected Euclidean distance is small enough, the difference between $\mathbb{E}[F(q, r)]$ and $F(q, r_c)$ can be controllable (or even close to zero).

Based on the lemmas we have:

**Theorem 1.** *Given $\mathbb{E}[F(q, r)] < \infty$, we have*

$$|F(q, R_q) - F(q, \hat{r}_c)| \xrightarrow{a.s.} 0 \qquad when \ |R_q| \to \infty \ and \ \mathbb{E}\|r - r_c\|_2 \to 0 \tag{13}$$

*where $\hat{r}_c$ is the estimation of $r_c$ based on the well-trained DBM.*

---

[3]The proof of Lemma 1, Lemma 2 and Theorem 1 is given in Appendix B.

***proof of theorem 1***.

$$|F(q, R_q) - F(q, \hat{r}_c)| \leq |F(q, R_q) - \mathbb{E}[F(q, r)]| + |\mathbb{E}[F(q, r)] - F(q, r_c)| + |F(q, r_c) - F(q, \hat{r}_c)| \tag{14}$$

Following lemma 1, the first term converges almost surely to zero when $|R_q| \rightarrow \infty$.

And following lemma 2, the second term goes to zero when $\mathbb{E}\|r - r_c\|_2 \rightarrow 0$.

Finally, given the well-trained DBM with the capability of mapping a query $q$ to $\hat{r}_c$ (Wang et al., 2010; Srivastava & Salakhutdinov, 2012), $\hat{r}_c$ can be taken as the estimation of $r_c$. Meanwhile, since the function $F(q, r)$ is the composition and combination of simple continuous functions, we have $|F(q, r_c) - F(q, \hat{r}_c)| \rightarrow 0$. □

## 4 THE HYBRID LOSS OF ADVERSARIAL RESPONSE GENERATION

As discussed in Section 2, the ability of the response generator in the general GAN architecture is limited to learning the dense distribution around $\frac{1}{|R_q|} \sum_{r \in R_q} D(q, r)$ (see Equation 5), which is composed of the most frequent patterns in the semantic space. By contrast, it is difficult for the general architecture to sense the remaining sparse space containing high-quality diverse responses. Therefore, reasonably describing the local distribution of the responses to a given query is highly necessary. According to the analysis in Section 3, the average free energy can be taken to model the state of the local area of the responses to a query, and such energy can be reasonably approximated via the DBM defined on the query-response pairs. On the basis of the previous sections, this section will finally propose the new hybrid loss function to force the generator to produce responses with better diversity through the more stable adversarial training process.

### 4.1 THE RADIAL DISTRIBUTION FUNCTION OF THE RESPONSE

The analysis in Section 3 have shown that the local distribution state of the responses $R_q$ to the given query $q$ can be modeled by the average free energy $F(q, R_q)$. On this basis, it is possible to propose the description of the spatial state of a single response $r$ in the semantic space, and consequently, we can give a new adversarial loss indicating the cost of simulating the local distribution of $R_q$.

According to Definition 3.1 and 3.2, in each response cluster $R_q$, the distance from a response $r$ to the cluster center $r_c$ is actually equivalent to the scaled energy difference between them, that is,

$$\Delta_{q,r,r_c} = \frac{F(q, r) - F(q, r_c)}{F(q, R_q)} \tag{15}$$

Meanwhile, on the basis of Theorem 1, $F(q, R_q)$ can be approximated by $F(q, \hat{r}_c)$, and $\hat{r}_c$ is modeled from training data, and thus we have:

$$\Delta_{q,r,r_c} \approx \frac{F(q, r) - F(q, \hat{r}_c)}{F(q, \hat{r}_c)} = \alpha_{(q,r)} \tag{16}$$

Here we approximate $\Delta_{q,r,r_c}$ with $\alpha_{(q,r)}$, and formally call $\alpha_{(q,r)}$ as the Radial Distribution Function (RDF), indicating the relative cost ratio to $F(q, r_c)$ for obtaining $r$ from a given $q$ (also the distinctiveness of $r$, actually).

### 4.2 THE HYBRID OBJECTIVE FUNCTION

Based on the previous discussions, for the adversarial response generation methodology, the essence is to reasonably describe the state of the local distribution of the response cluster given by Definition 3.2, and further more, to take this important element into account in the final optimization.

Especially, in Subsection 4.1, we have defined the Radial Distribution Function in Equation 16 to quantify the distinctiveness of a response, the very basis of which is the description of the local state $F(q, R_q)$ in the semantic space. Thus, we can further build a mechanism to quantify the difference between the generated response and the golden response as follows:

$$\delta\alpha = \alpha_{(q,r)} - \alpha_{(q,\tilde{r})} \tag{17}$$

where $\tilde{r}$ is the generated response given by the generator and $r$ comes from the original data. If $\delta\alpha$ moves toward zero, $\alpha_{(q,\tilde{r})}$ would be close to $\alpha_{(q,r)}$ sharing the same $F(q, r_c)$.

Consequently, a new expectation comes out. That is, the generator needs to provide results that can minimize $\delta\alpha$, so as to fit the local distribution of the existing responses to a given query. Thus, a hybrid objective of the generator can be finally defined as:

$$\min J^G = -\mathbb{E}\left[\log D(q, r)\right] + ReLU(\delta\alpha) \tag{18}$$

A hinge loss, conducted by the $ReLU$ function $ReLU(\delta\alpha) = \max(0, \ \delta\alpha)$, is especially introduced to reform $\delta\alpha$. The primary reason of this operation is that the $ReLU$ function has positive output only if $\delta\alpha \geqslant 0$, according to the definition of $ReLU(\delta\alpha)$. Apparently, $\delta\alpha < 0$ indicates that the generated response $\tilde{r}$ is too far from the center of the response cluster in the semantic space, so that its relevance may be highly questionable. Meanwhile, minimizing a negative variable is against the optimization direction. After the ReLU transformation, there remains valid loss only when $\delta\alpha \geqslant 0$, and thus both the diversity and the relevance of generated results are taken into account.

## 4.3 THE PHASE-WISE OPTIMIZATION

According to the analysis in Section 2, the trivial adversarial training directed by $-\mathbb{E}\left[\log D(q, r)\right]$ can only determine the form of general responses to a given query. From the spatial perspective in the semantic space, the original adversarial objective is helpful to roughly locate the response cluster to be generated. However, the local distribution can not be captured by this procedure.

By contrast, according to the discussions above, the proposed hybrid objective actually provides a way to force the generated responses, originally gathering around the general form, to expand into the expected local shape described by the golden truth, by conducting a phase-wise optimizing operation. This mechanism can be detailed in an intuitive way:

***Foundation***: Once the DBM in Figure 1 is well-trained with the query-response corpus, the semantic center $r_c$ of a Response Cluster can be determined by the given query $q$.

***Phase-1***: In the early stage of the adversarial training, a generated response $\tilde{r}$ is not semantically relevant to the query $q$. Thus, it can be inferred that $\tilde{r}$ is radially farther from the cluster center $r_c$ than the golden response $r$. In this situation, according to Equation 16 and Equation 17, we can claim that $\delta\alpha \leq 0$. In this phase, the hyper objective goes back to the general adversarial objective due to the ReLU function. Thus, the model is trying to force the generated samples to approach the center of each cluster, ignoring local distributions.

***Phase-2***: During the adversarial training in *Phase-1*, the generated result $\tilde{r}$ will go approaching to the cluster center $r_c$, which means $\alpha_{(q,\tilde{r})} \rightarrow 0$ . It should be noted that, for any meaningful existing training sample $r$, $\alpha_{(q,r)} > 0$. Therefore, at some point, it turns to $\delta\alpha > 0$ and the right part of the hybrid objective in Equation 18 takes effect. Consequently, for each given query, the distribution of the generated results will expand to fit the local distribution of the golden samples.

## 5 EXPERIMENTS

### 5.1 EXPERIMENTAL SETUPS

**Datasets.** Our experiments are conducted on two main stream open-access conversation corpora: The Opensubtitles corpus and the Sina Weibo corpus. The OpenSubtitles dataset contains 5,200,000 movie dialogues, where we extract query-response pairs following (Xu et al., 2018; Li et al., 2016). The Sina Weibo Corpus (Shang et al., 2015a) contains 2,500,000 single-turn Chinese dialogues, in which the length of the query and response ranges from 4 to 30. We sample 100,000, and 2,000 unique query-response pairs as validation and testing dataset respectively from both of the corpora[4].

**Baselines.** For meaningful comparison, we introduce the following models as the baselines:

(1) ***Seq2Seq***: a sequence-to-sequence model trained with maximum likelihood estimation (MLE);

---

[4]Both the English and the Chinese datasets used in our experiments are uploaded to `https://www.dropbox.com/sh/k8i079gd2111lsb/AACLLtlNAzile543Da8Qs9tFa?dl=0`.

Table 1: Performances of LocalGAN and Baselines on the Opensubtitles and Weibo Datasets.

| Model | Opensubtitle | | | | Weibo | | | |
|---|---|---|---|---|---|---|---|---|
| | Dist-1 | Dist-2 | Ent4 | Rel. | Dist-1 | Dist-2 | Ent4 | Rel. |
| Seq2Seq | 0.025 | 0.081 | 5.650 | 1.090 | 0.055 | 0.153 | 6.400 | 0.315 |
| Seq2Seq-MMI | 0.027 | 0.086 | 5.698 | 1.067 | 0.059 | 0.172 | 6.860 | 0.309 |
| Adver-REGS | 0.0296 | 0.098 | 5.701 | 1.113 | 0.061 | 0.181 | 7.658 | 0.320 |
| GAN-AEL | 0.030 | 0.100 | 5.733 | 1.106 | 0.062 | 0.183 | 7.765 | 0.318 |
| AIM | 0.0292 | 0.095 | 5.783 | 1.120 | 0.064 | 0.189 | 7.833 | 0.321 |
| DAIM | 0.031 | 0.103 | 5.873 | 1.098 | 0.067 | 0.195 | 8.042 | 0.316 |
| LocalGAN | **0.036** | **0.110** | **6.073** | **1.132** | **0.071** | **0.212** | **8.561** | **0.327** |

(2) **Seq2Seq-MMI**: the NRG model with a Maximum Mutual Information criterion (Li et al., 2016);

(3) **Adver-REGS**: the NRG model trained using adversarial framework, in which the policy gradient was employed to transfer the reward of the discriminator to the generator (Li et al., 2017);

(4) **GAN-AEL**: an adversarial framework with an approximate embedding layer for connecting the generator with the discriminator directly (Xu et al., 2017).

(5) **AIM / DAIM**: the adversarial training strategy allowing distributional matching of synthetic and real responses and explicitly optimizing a variational lower bound on pairwise mutual information between the query and response, so as to improve the informativeness and diversity of generated responses (Zhang et al., 2018b)[5].

**Evaluation Metrics.** To evaluate diversity of results, we adopt three widely-applied metrics: Distinct-1 (**Dist-1**), Distinct-2 (**Dist-2**), and Entropy (**Ent4**) (Li et al., 2016; Zhang et al., 2018b; Jost, 2006). Besides, the relevance (**Rel.**) is measured by summing three embedding-based similarities (greedy, average, extreme) (Liu et al., 2016) upon the ground-truth and generated responses.

**Training Details.** The vocabulary size of both datasets is 40,000. The embedding layer of OpenSubtitles and Sina Weibo is initialized using 200-dimensional Glove vectors (Pennington et al., 2014) and 300-dimensional Weibo vectors (Li et al., 2018) respectively. All the models are first pre-trained by MLE, and then the models including Adver-REGS, GAN-AEL, AIM, DAIM and LocalGAN are trained with adversarial learning. The discriminator of Adver-REGS and GAN-AEL are based on CNN following (Yu et al., 2017; Xu et al., 2017), in which the filter sizes are set to (1,2,3,4) and the filter number is 128, while that of LocalGAN adopts DBM with ($2\times$embedding_size, 128, 128) to represent the semantic of queries and responses. The hidden size of the generator is set to 256 and 512 in GAN-based models and Seq2Seq respectively. To guarantee the performance consistency of AIM and DAIM, we adopt the recommended parameter settings given by Zhang et al. (2018b). The experiments are conducted on the Tesla K80 GPU.

## 5.2 RESULTS & ANALYSIS

Table 1 lists quantitative results on the diversity and relevance of generated responses on both datasets. As shown by the results, compared to Seq2Seq and Seq2Seq-MMI, the GAN-based methods give better results on the diversity oriented metrics, including Dist-1, Dist-2 and Ent4. This observation indicates that adversarial learning does provide the meaningful guidance to NRG models to avoid some of the safe-responses.

It can be observed that LocalGAN outperforms the baselines with adversarial learning architecture (Adver-REGS, GAN-AEL, AIM and DAIM) on both the diversity metrics and the relevance metrics. Generally, a notable improvement on diversity may lead to some negative influence on relevance, and thus promoting the diversity of generated response while maintaining their relevance is essentially desired for any methodologies, which has been achieved by our LocalGAN. The performances of LocalGAN can be attributed to the fact that LocalGAN has taken the local distribution of responses to a given query into account. By adopting the hybrid objective function, the proposed adversarial

---

[5]We have taken the codes of AIM and DAIM from `https://github.com/dreasysnail/converse_GAN` implemented by the authors of this work for comparisons.

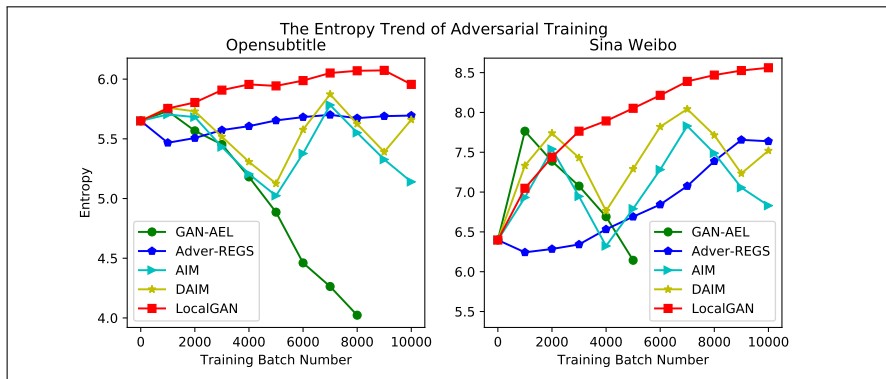

Figure 2: The Entropy Trend of adversarial learning based models in the Training Process.

model gets to capture the spatial characteristics of response clusters, and the generation process is consequently forced to fit the semantic distributions of response clusters.

The training stability is a tough issue to be addressed for adversarial learning (Yu et al., 2017), and as discussed in the previous sections, one of the motivations of our LocalGAN is to make adversarial learning more stable. To verify this aspect, we track the changing of the Entropy (Ent4) of results given by GAN-AEL, Adver-REGS, AIM, DAIM and LocalGAN, as shown in Figure 2.

It can be observed that the training of LocalGAN and Adver-REGS is stable. By contrast, there exist obvious fluctuations on the curves of AIM and DAIM, and GAN-AEL rapidly gets out of control after 1000 batch, which makes it rather difficult to grasp the models with the best status. This group of results indicates the necessity of introducing additional restrictions into adversarial learning processes. For this purpose, Adver-REGS introduces a teacher-forcing loss (Li et al., 2017), while AIM and DAIM have taken the informativeness oriented constraints to partially control the stability (Zhang et al., 2018b). However, GAN-AEL only takes the Wasserstein distance as the objective (Xu et al., 2017), and thus the entropy goes down rapidly. Compared to the Adver-REGS, our LocalGAN achieves better diversity with even a more smooth entropy curve. The training of LocalGAN benefits from the phase-wise optimization driven by the hybrid loss, and its stability also indicates the meaningfulness of modeling and utilizing local distributions of responses.

## 5.3 HUMAN EVALUATION

To further conduct intuitive comparisons among the NRG models, we perform human evaluations on 500 testing samples. Five annotators are asked to judge whether a response is relevant to the given query and whether the response is informative or not respectively. Both human metrics "Relevance" and "Informativeness" solely have two labels 0 and 1 (0: irrelevant or non-informative; 1: relevant or informative) for simplicity. The human evaluation results are listed in Table 2, and the results basically match the observations in Table 1. The Kappa of relevance and informativeness annotations are 0.734 and 0.561 respectively.

| Method | Relevance | Informativeness |
|---|---|---|
| Seq2Seq | 0.738 | 0.25 |
| Seq2Seq-MMI | 0.67 | 0.336 |
| Adver-REGS | 0.702 | 0.398 |
| GAN-AEL | 0.696 | 0.41 |
| AIM | 0.746 | 0.294 |
| DAIM | 0.768 | 0.45 |
| LocalGAN | **0.784** | **0.536** |

Table 2: Human Evaluation Results of Models.

## 5.4 CASE STUDY

The effectiveness of the NRG models can be illustrated by the responses generated by them intuitively. Table 3 lists the sampled results given by all the baselines and our proposed LocalGAN, trained on the OpenSubtitle corpus (The samples based on the Sina Weibo dataset are given in Appendix C). It can be observed that, compared to the basic Seq2Seq and Seq2Seq-MMI, the models incorporating adversarial learning have generated results with varying degrees of semantic diversity,

as expected. Among the baselines, DAIM produces more diverse responses, which matches the quantified evaluation results in Table 1, The samples from LocalGAN have shown better semantic relevance, and meanwhile the diversity and informativeness are also guaranteed.

Table 3: Sample results given by different models trained on the OpenSubtitle corpus.

| | |
|---|---|
| **Query:** you know you're a strong woman and i admire that. 
 **Ground-truth:** i'm sorry for what i'm about to do. | **Query:** you are a perfect girlfriend. 

 **Ground-truth:** some times i wish that you would go out more. |
| **Seq2Seq:** you know what i mean. 
 **Seq2Seq-MMI:** you know what you're doing. | **Seq2Seq:** what are you talking about? 
 **Seq2Seq-MMI:** you have no idea what you're talking about. |
| **Adver-REGS:** i don't know how to do that. 
 **GAN-AEL:** what are you talking about? 
 **AIM:** what are you doing here? 
 **DAIM:** i had a plan for you. | **Adver-REGS:** do you have a girlfriend? 
 **GAN-AEL:** what are you doing here? 
 **AIM:** you are the only one. 
 **DAIM:** you are a good person. |
| **LocalGAN:** i thought it was your idea. | **LocalGAN:** you are a part of me. |
| **Query:** maybe cause you keep marrying for money. 
 **Ground-truth:** i did not marry for money. | **Query:** you're just my roommate eric. 

 **Ground-truth:** millie is much much more than that. |
| **Seq2Seq:** what do you mean? 
 **Seq2Seq-MMI:** what do you want me to do? | **Seq2Seq:** i don't know what you're talking about. 
 **Seq2Seq-MMI:** i need to talk to you. |
| **Adver-REGS:** what do you want to say? 
 **GAN-AEL:** what is the difference? 
 **AIM:** what are you talking about? 
 **DAIM:** what is the problem? | **Adver-REGS:** i want to talk to you about this. 
 **GAN-AEL:** can you help me please? 
 **AIM:** what are you talking about. 
 **DAIM:** its a pleasure to meet you. |
| **LocalGAN:** i don't have a choice. | **LocalGAN:** no you are my friend. |
| **Query:** he says some kind of sea creatures washed up on his beach. 
 **Ground-truth:** how could it be a sea creature? | **Query:** would you excuse us for just a minute? 

 **Ground-truth:** ted i'm gonna be honest. |
| **Seq2Seq:** i don't know what to do. 
 **Seq2Seq-MMI:** what are we supposed to do? | **Seq2Seq:** why are you doing this? 
 **Seq2Seq-MMI:** i don't want to see you again. |
| **Adver-REGS:** he wants to go to the sandglass. 
 **GAN-AEL:** he told us to come back. 
 **AIM:** i thought what was going to happen. 
 **DAIM:** he said it was a gift. | **Adver-REGS:** what are you going to do? 
 **GAN-AEL:** i have to go to the bathroom. 
 **AIM:** what are you supposed to do? 
 **DAIM:** why are you calling me? |
| **LocalGAN:** i thought it looks like a fish. | **LocalGAN:** please give me your reason for absence. |

## 6 CONCLUSIONS

This paper has given the theoretical proof of the upper bound of the adversarial training leveraged models on the Seq2Seq-based neural response generation task. The proof indicates that, due to the local distribution nature of query-response corpora, the GAN based NRG models will converge to the states mostly generating specialized patterns corresponding to given queries. To address this issue, we proposed to model the local distribution of queries and their response in the semantic space by adopting energy-based function, and found the approximation of this function. According to this approximated distribution representation, a new loss function describing the local expansion cost in the fitting of response distribution is presented and finally combined with the traditional GAN loss to form a hybrid training objective for the GAN based NRG model. This paper provides a reasonable explanation to the unstable training process and unsatisfying results of GAN based NRG approaches, and meanwhile gives a different perspective to leverage the local data distribution to enhance classic GAN approaches.

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

## A  THE GRAPHICAL MODEL FOR RESPONSE DISTRIBUTION MODELING

Different from the computer vision related scenario, training a DBM on a set of text vectors is not trivial, since the training procedure is difficult to converge due to the value scale of text vectors is much larger than image vectors. Moreover, the simple scaling methods are not effective enough for this issue. For this purpose, this paper adopts standard-scaler[6] to remove the mean and scale to unit variance. To valid the effectiveness of standard-scaler, We conduct experiments on the query-response matching task using normalized vectors. The experiment result show that the matching performance based normalized vectors is similar to that of CNN based architecture (Kim, 2014).

## B  DETAILED PROOF OF LEMMAS

***Proof of Lemma 1***. As mentioned before, $r_i$ is assumed to be independent and identically distributed (i.i.d) normal variable, thus $F(q, r_i)$ can be seen as i.i.d random variable for a fixed query $q$. Based on the strong law of large numbers,

$$F(q, R_q) = \frac{1}{|R_q|} \sum_{r_i \in R_q} F(q, r_i) \xrightarrow{a.s.} \mathbb{E}[F(q, r)] \qquad when \; |R_q| \to \infty$$

$\square$

***Proof of Lemma 2***. For a fixed query, $F(q, r)$ can be seen as the the scalar function of vector $r$. For simplicity, we denote $F(q, r)$ as $f(r)$.

Taylor expansions for the first moment of function of random variables are as follows.

$$\mathbb{E}[f(r)]] = \mathbb{E}\left[f\left(r_c + (r - r_c)\right)\right]$$
$$= \mathbb{E}\left[f(r_c) + (r - r_c)^T Df(r_c) + \frac{1}{2}(r - r_c)^\top \left\{D^2 f(r_c)\right\}(r - r_c) + R_{r_c,2}(r - r_c)\right]$$

where $Df(r_c)$ is the gradient of $f$ evaluated at $r_c$, $D^2 f(r_c)$ is the Hessian matrix and $R_{r_c,2}(r - r_c)$ is the Lagrange remainder. Since $\mathbb{E}r = r_c$, the second term $(r - r_c)^T Df(r_c)$ disappears.

Then we will try to find the upper bound of the third term and the remainder term. The relevant theorems used in the proof are listed as follows.

- According to (Petersen & Pedersen, 2012), assume A is symmetric, $c = \mathbb{E}[\mathbf{x}]$ and $\mathbf{\Sigma} = \mathrm{Var}[x]$, then
$$\mathbb{E}\left[x^T A x\right] = \mathrm{Tr}(A\mathbf{\Sigma}) + c^T A c.$$

- (Mirsky, 1975) states following theorem: If $A, B$ are complex $n \times n$ matrices with singular values $\alpha_1 \geq \alpha_2 \geq \cdots \geq \alpha_n$ and $\beta_1 \geq \beta_2 \geq \cdots \geq \beta_n$ respectively, then
$$|\mathrm{Tr}(AB)| \leq \sum_{i=1}^n \alpha_i \beta_i$$

Firstly, since $r - r_c \sim N(0, \Sigma)$ and $\Sigma$ is positive semi-definite matrix, the third term can be simplified as follows.

$$|\mathbb{E}[(r - r_c)^\top \left\{D^2 f(r_c)\right\}(r - r_c)]| = |\mathrm{Tr}(\{D^2 f(r_c)\}\Sigma) + \mathbf{0}^\top D^2 f(r_c)\mathbf{0}|$$
$$\leq \sum_{i=1}^n \alpha_i \beta_i$$
$$\leq \alpha_1 \mathrm{Tr}(\Sigma) = \alpha_1 \mathbb{E}\left[\|r - r_c\|_2^2\right]$$

where $\alpha_1 \geq \alpha_2 \geq \cdots \geq \alpha_n$ and $\beta_1 \geq \beta_2 \geq \cdots \geq \beta_n$ denote the singular value of matrix $\{D^2 f(r_c)\}$ and $\Sigma$ respectively.

---

[6]https://scikit-learn.org/stable/modules/generated/sklearn.preprocessing.StandardScaler.html

Meanwhile, according to the Corollary 1 in (Folland, 2005), we have that

$$|R_{r_c,2}(r - r_c)| \leq \frac{\tilde{M}}{3!} \|r - r_c\|_1^3$$

where $\tilde{M}$ is the upper bound for absolute value of third-order partial derivatives of $f$.

Next, we show that the $\alpha_1$ and third-oder partial derivatives can be bounded by $M$, where $M$ is $\max_{h_q, 1 \leq i \leq n} |b(h_q, i)|$ ($h_q$ follows multinomial distribution) and $n$ is the dimension of $r$.

Substituting the definition of $E(q, r, H)$ into $F(q, r)$, we have following equation:

$$f(r) = F(q, r) = -\log \sum_{h_q, h} exp(h_q^T W_{qr} r + h_q^T W_{qq} q + h^T W_{qh} h_q).$$

After that, its first-order, second-order and third-order partial derivative are calculated as follows:

$$\frac{\partial f(r)}{\partial r_i} = -\frac{\sum_{h_q, h} exp(h_q^T W_{qr} r + h_q^T W_{qq} q + h^T W_{qh} h_q) \times (h_q^T W_{qr})_i}{\sum_{h_q, h} exp(h_q^T W_{qr} r + h_q^T W_{qq} q + h^T W_{qh} h_q)}$$

$$\frac{\partial^2 f(r)}{\partial r_i \partial r_j} = \sum_{h_q, h} a(h_q, h) \times b(h_q, i) \times \left[ \sum_{\tilde{h}_q, \tilde{h}} a(\tilde{h}_q, \tilde{h}) \times b(\tilde{h}_q, j) - b(h_q, j) \right]$$

$$\frac{\partial^3 f(r)}{\partial r_i \partial r_j \partial r_k} = \sum_{h_q, h} c(h_q, h, k) \times b(h_q, i) \times \left[ \sum_{\tilde{h}_q, \tilde{h}} a(\tilde{h}_q, \tilde{h}) \times b(\tilde{h}_q, j) - b(h_q, j) \right]$$

$$+ \sum_{h_q, h} a(h_q, h) \times b(h_q, i) \times \left[ \sum_{\tilde{h}_q, \tilde{h}} c(\tilde{h}_q, \tilde{h}, k) \times b(\tilde{h}_q, j) \right]$$

where

$$a(h_q, h) = \frac{exp(h_q^T W_{qr} r + h_q^T W_{qq} q + h^T W_{qh} h_q)}{\sum_{\tilde{h}_q \tilde{h}} exp(\tilde{h}_q^T W_{qr} r + \tilde{h}_q^T W_{qq} q + \tilde{h}^T W_{qh} \tilde{h}_q)}$$

$$c(h_q, h, k) = \frac{\partial a(h_q, h)}{\partial r_k} = a(h_q, h) b(h_q, k) - a(h_q, h) \sum_{\tilde{h}_q, \tilde{h}} a(\tilde{h}_q, \tilde{h}) b(\tilde{h}_q, k)$$

and $b(h_q, i) = (h_q^T W_{qr})_i$ representing the $i$-th element of the vector $h_q^T W_{qr}$. According to the definition of $a(h_q, h)$, it is obvious that $a(h_q, h) > 0$ and $\sum_{h_q, h} a(h_q, h) = 1$.

The upper bounds for the second-order and third-order partial derivative are shown as follows.

$$\left| \frac{\partial^2 f}{\partial r_i \partial r_j} \right| \leq \sum_{h_q, h} |a(h_q, h)| \times |b(h_q, i)| \times \left[ \sum_{\tilde{h}_q, \tilde{h}} \left| a(\tilde{h}_q, \tilde{h}) \times b(\tilde{h}_q, j) \right| + |b(h_q, j)| \right] \leq 2M^2$$

$$|c(h_q, h, k)| \leq |a(h_q, h)| \times |b(h_q, k)| + |a(h_q, h)| \times \sum_{\tilde{h}_q, \tilde{h}} |a(\tilde{h}_q, \tilde{h})| \times |b(\tilde{h}_q, k)| \leq 2M|a(h_q, h)|$$

$$|\frac{\partial^3 f(r)}{\partial r_i \partial r_j \partial r_k}| \leq \sum_{h_q, h} |c(h_q, h, k)| \times M \times [M + M] + M^2 \sum_{\tilde{h}_q, \tilde{h}} |c(\tilde{h}_q, \tilde{h}, k)| \leq 6M^3$$

Based on the upper bound above, we can see that $\tilde{M}$ can be $6M^3$ and the upper bounds for $\alpha_1$ and $|R_{r_c,2}(r - r_c)|$ are as follows.

$$\alpha_1 = \sigma_{max}(\{D^2 f(r_c)\}) \leq \|\{D^2 f(r_c)\}\|_F = \sqrt{\sum_{i=1}^n \sum_{j=1}^n |\frac{\partial^2 f}{\partial r_i \partial r_j}|^2} \leq 2nM^2$$

$$|R_{r_c,2}(r - r_c)| \leq \frac{\tilde{M}}{3!}\|r - r_c\|_1^3 = M^3\|r - r_c\|_1^3 \leq (\sqrt{n}M\|r - r_c\|_2)^3$$

Hence, we have

$$
\begin{aligned}
|\mathbb{E}[f(r)] - f(r_c)| &= |\frac{1}{2}\mathbb{E}\big[(r - r_c)^\top \{D^2 f(r_c)\}(r - r_c)\big] + \mathbb{E}\big[R_{r_c,2}(r - r_c)\big]| \\
&\leq \frac{1}{2}\alpha_1\mathbb{E}\|r - r_c\|_2^2 + \mathbb{E}|R_{r_c,2}(r - r_c)| \\
&\leq \mathbb{E}(\sqrt{n}M\|r - r_c\|_2)^2 + \mathbb{E}(\sqrt{n}M\|r - r_c\|_2)^3
\end{aligned}
$$

Therefore,

$$\lim_{\mathbb{E}\|r - r_c\|_2 \to 0} |\mathbb{E}[f(r)] - f(r_c)| = 0$$

$\square$

## C ADDITIONAL CASES

| | |
|---|---|
| **Query:** 凡事不要想的太复杂，手握的太紧，东西会碎，手会疼。
You don't need to make things too complicated, and it's
just like something you are holding, if you use too much power,
it will break and hurt you.
**Ground-truth:** 没办法啊。为什么我那么敏感呢？
I feel very helpless. Why am I so sensitive? | **Query:** 河豚君。生气的样子真呀么真可爱。
The globefish looks so cute when it gets angry.


**Ground-truth:** 我还以为他吞了一个球呢。
I thought it had swallowed up a ball. |
| **Seq2Seq:** 这句话说的很好！
That was spoken well.
**Seq2Seq-MMI:** 这句话说得太好了。
That was spoken quite well. | **Seq2Seq:** 我也是这样的啊。
So am I.
**Seq2Seq-MMI:** 原来是这样的啊！
So that's it! |
| **Adver-REGS:** 这句话很有道理。
It quite make sense.
**GAN-AEL:** 这句话是什么意思？
What does that mean?
**AIM:** 这句话说得太对了！
These words speak the truth.
**DAIM:** 那是我自己想要的东西。
That's what I want. | **Adver-REGS:** 我也有一个这样的。
I also have one like this.
**GAN-AEL:** 我也有这样的感觉。
I also have the same feeling.
**AIM:** 是啊，我也是这样的。
Yeah, so am I.
**DAIM:** 好可爱的样子！
It looks so cute! |
| **LocalGAN:** 放下，放下，立地成佛。
Let it go and free yourself. | **LocalGAN:** 这就是传说中的河豚？
So this is the legendary globefish? |
| **Query:** 给所有的胖纸讲个鬼故事，夏天来了。
A terrible story to all the fatties: the summer is coming.
**Ground-truth:** 现在也就这只肥猫能让我笑一笑。
This fatty cat looks so funny. | **Query:** 远离城市，才得安宁。
You can feel peaceful only if you are far from cities.
**Ground-truth:** 我也是这么想的，但真正有几个人能做到呢。
Agree with that, but I think few people can really do it. |
| **Seq2Seq:** 我只想说这句话我很喜欢
I just wanna to say that I like this sentence.
**Seq2Seq-MMI:** 我只想说这句话是什么意思？
I just wanna say what do you mean? | **Seq2Seq:** 我也是这么认为
I also think so .
**Seq2Seq-MMI:** 我也有这种感觉
I have the same feeling as well. |
| **Adver-REGS:** 我只能说，你是我的骄傲
I can only say that I am proud of you.
**GAN-AEL:** 我不会也有这样的一天吧。
May I have such a day ?
**AIM:** 我只能说，我也是这样的。
I can only say so do I.
**DAIM:** 你是我的唯一，我的瘦子。
You are my only, skinny. | **Adver-REGS:** 这是什么意思？
What does this mean?
**GAN-AEL:** 又不是一个人的城市。
This is not a city for any single person.
**AIM:** 我也想，但我不会去。
I also want it, but I will not take action.
**DAIM:** 这就是我们的生活！
This is our life! |
| **LocalGAN:** 这就是传说中的瘦子的心声。
This must come from the heart of a skinny guy. | **LocalGAN:** 心若不静，耳根亦不清静。
If you don't get inner peace, you will hear the boring noise. |
| **Query:** 只要生命还在，就可以从头再来。
Never give up as long as you are alive.

**Ground-truth:** 有些东西. 不能重来。
But there always exist something not repeatable. | **Query:** 喜欢这样被婴儿的小手抓住手指的童鞋举手！
Just let me know if you also love to be holding on
your finger by the tiny hand of a baby!
**Ground-truth:** 喜欢那带着稚气声音喊我姐姐、娜姨
I love to be called as the sister or auntie with baby voice. |
| **Seq2Seq:** 加油，永远支持你！
Cheer up, and I am always supporting you.
**Seq2Seq-MMI:** 每一天都是最好的。
Every day is the best day. | **Seq2Seq:** 我也想要这样的！
I also want this!
**Seq2Seq-MMI:** 我喜欢这种感觉！
I love this feeling!. |
| **Adver-REGS:** 新的一天，加油！
A new day, cheer up!
**GAN-AEL:** 新的一天，新的起点！
New day, new beginning!
**AIM:** 加油，继续加油！
Cheer up, forever!
**DAIM:** 一切都会过去的。
Everything will pass. | **Adver-REGS:** 我家也是这样的！
The one in my home is like this as well!
**GAN-AEL:** 有木有想要一个？
Do you want one like this?
**AIM:** 好喜欢这种感觉。
Love this feeling so much.
**DAIM:** 我也想要一个这样的男朋友
I also want a boyfriend like this. |
| **LocalGAN:** 淡定吧，一切都会好起来的。
Be calm, and everything will be OK. | **LocalGAN:** 好有爱啊，我也想要一个宝宝。
So cute, I also want a baby. |

Figure 3: Sample Results given by different models trained on the Sina Weibo corpus.

