# OpenReview forum: "LocalGAN: Modeling Local Distributions for Adversarial Response Generation"
_ICLR.cc/2020/Conference — Reject_

### Official Review · AnonReviewer2 · 2019-10-21
**Official Blind Review #2**

**Rating:** 3

**Review:**

Contributions:

The main contribution of this paper lies in the proposed LocalGAN for neural response generation. The key observation is that for a given query, there always exists a group of diverse responses that are reasonable, rather than a single ground-truth response. Therefore, the local semantic distribution of responses given a query should be modeled. Besides the original GAN loss, the proposed LocalGAN adds an additional local-distribution-oriented objective, resulting in a hybrid loss for training, which claims to achieve better performance on response generation datasets.

Strengths:

I think the proposed model contains some good intuitions, that is, the generated responses should be modeled as a local distribution, rather than a single ground-truth output during training. The motivation of this paper is therefore clear. Experimental results in Table 1 seems encouraging.

However, I would have to say that the current draft is poorly presented. There are a lot of unclear parts that should be more carefully clarified, with details below.

Weaknesses:

(1) Writing: I think the language in this paper is repetitive, and can be much more precise and concise. Also, there are typos here and there throughout the whole draft. I would suggest the authors doing a careful proofreading before next submission.

Minor: in the line before Eqn. (4), change "SIMPLY" to "simply".

(2) Clarity: Overall, the presented method is unclear.

a) It is not entirely clear what the authors mean by saying "this paper has given the theoretical proof of the upper-bound of the adversarial training ...". I am not sure whether Eqn. (6) is totally correct, or at least how useful it is.
b) The notations throughout the paper is a little bit confusing. The authors should normalize all the notations to be consistent.
c) It is not clear what Eqn. (3) truly means. What is the value for s? The KL divergence should take two distributions as input, but here, the input are two triplets.
d) In the line below Eqn. (6), what is \tilde{R}_q? This is not defined.
e) The proposed method relies on the use of R_q. However, how to define, or learn R_q is not clear. In the dataset, given a given query q, how do we find R_q?
f) It is not clear why Deep Boltzmann Machines are needed here. I'd like the authors to more clearly clarify this design. Further, since DBM is used, then how the final model is trained together? Now, the models contains both adversarial learning, and contrastive-divergence-based algorithms for DBM training. This seems make the whole model training more unstable.
g) Generally, I think Section 3 and Section 4 are hard to follow. Further, I did not see how useful Lemma 1 & 2 and Theorem 1 are. The final objective Eqn. (17) is also confusing.

(3) Experiments: My biggest concern about the experiments is that human evaluation should be conducted, given the subjective nature of the task. This is lacked in the current draft. Only reporting numbers like Table 1 is not convincing.

** This paper provides a link that actually links to a github repo. I am not sure whether this violates the policy of ICLR submissions or not. But at least from my point of review, this link should be anonymized. **

**Experience Assessment:**

I have published one or two papers in this area.

**Review Assessment: Checking Correctness Of Derivations And Theory:**

I assessed the sensibility of the derivations and theory.

**Review Assessment: Checking Correctness Of Experiments:**

I assessed the sensibility of the experiments.

**Review Assessment: Thoroughness In Paper Reading:**

I read the paper at least twice and used my best judgement in assessing the paper.

---

> ### Author Response · Authors · 2019-11-11
> **Responses to Reviewer #2**
>
> We are grateful for your valuable comments, and the responses to your concerns are given as follows.
>
> 1.	Writing: We have conducted a proofreading upon our draft carefully, with the typos and imprecise expressions revised.
>
> 2.	Clarity:
> 	a.	Eq. (6) (Eq. 5 in the final version) declares the upper bound of the adversarial learning based response generation, defined by \sum_{q} \mathbb{E}_{(q, r) \thicksim {q, R_q}}\left[ \log D(q, r)\right]. Actually, it is the extended version of Inequation (5) (Inequation 4 in the final version) by taking the response group R_q into consideration rather than a single response, which is the very motivation of our paper. As mentioned in Page 3, Eq. (6) (Eq. 5 in the final version) is given based on Eq. (4) and Ineq .(5) (Eq. 3 and Ineq. 4 in the final version), and both of them are the basis of the general GAN.
> Eq. (6) (Eq. 5 in the final version) indicates that, actually, optimization process of adversarial learning upon conversational datasets will make the generated responses converge to the center of each local distribution of R_q corresponding to each given dependent query, and the center is defined by \frac{1}{|R_q|}  \sum_{r\in R_q}D(q,r). This is apparently not expected and our work tries to give a solution to this issue.
>
> 	b.	We have checked the notations and modified the confusing ones.
>
> 	c.	In Eq. (3) (in the Original version), s stands for the confidence score given by the discriminator of GAN, which is highly relative to the capability of a GAN-based NRG model. Eq. (3) aims at ideally giving a formalized definition to the capability of the GAN model.  However, as you have suggested, this equation is not rigorous, and we also find it is not very necessary to involve an equation to explain our points, so we have deleted this equation in the revised version.
>
> 	d.	\tilde{R}_q is not included in Eq. (6) (Eq. 5 in the final version) actually, and we have modified the description under Eq. (6). Thanks for the reminding.
>
> 	e.	R_q ={r_q_i} stands for the finite set of the corresponding responses to the query q, as mentioned in Page 3. In fact, all the query-response corpora for NRG can be naturally organized into the form {q, R_q | R_q ={r_q_i}}, since the query q can be taken as the key of the item (q, r_q). That is, once a query-response dataset is prepared, R_q can be obtained easily by going through the dataset and reorganize it with a dictionary structure. We feel sorry about the confusion caused by the absent details and will complement them in the revised version.
>
> 	f.	DBM is used to model the local distribution state of the Response Cluster R_q, from the free energy based perspective (Eq. 8-10 in the original version, also Eq. 7-9 in the final version). Further, the free energy is used to define Radial Distribution Function (Eq. 14-15 in the original version, also Eq. 15-16 in the final version), which is highly relevant to the second part of the hybrid objective function (Eq. 17, also Eq. 18 in the final version).
> It should be noted that, the DBM is Pre-trained with the query-response coprus, and the training process is independent with the training of GAN. Thus, DBM does not need to be trained together with GAN, and will not make the GAN training unstable.
>
> 	g.	By proposing Lemma 1&2 and Theorem 1, we try to give a reasonable approximation of F(q, R_q), which is the very basis of the final hybrid objective as explained in item f above. In detail, Lemma 1 approximates F(q, R_q) with the expectation of F(q, r), denoted with \mathbb E [F(q,r)]. Lemma 2 further proves that \mathbb E [F(q,r)] can be approximated by F(q, r_c), where r_c stands for the center of the response cluster in the semantic space. On the basis of Lemma 1&2, Theorem 1 gives the final approximation of F(q, R_q).
> Theorem 1 leads to an executable loss function to quantify the cost of fitting the local distribution of the response cluster.
> As for the final objective Eq. 17 (Eq. 18 in the final version), Subsection 4.3 actually details its working phases during the training process.
>
> 3.	Experiments: Following the suggestion, we have conducted the human evaluation, and the results are given in Subsection 5.3 in the revised version.
>
> ** As for the Github Link of our code, we need to clarify that all the codes and the link itself have been already anonymized carefully. And the term Kramgasse49 in the link is actually the address of Einstein's House which means nothing to our code. The shared datasets are anonymized as well. **

---

### Official Review · AnonReviewer1 · 2019-10-23
**Official Blind Review #1**

**Rating:** 3

**Review:**

POST-REBUTTAL FEEDBACK

I share the same concerns as that of reviewer 2 in the response to the rebuttal. Hence, my score remains unchanged.


SUMMARY OF REVIEW

This paper motivates the need to "contextualize" responses based on the query to bring about stable training in NRG and consequently proposes localGAN to realize this. On the overall, I like the motivation and the proposed approach of this paper. The experimental results also look convincing.

On the flip side, the technical formulation and theoretical results are not presented rigorously and important technical details are missing, as discussed below. As a result, clarifications from the authors are needed to ensure the correctness of their formulation. The authors also need to improve the presentation and proof of the theoretical results; the correctness has to be checked again. In my opinion, these theoretical results do not improve my current assessment of the paper and can be removed to cut down to 8 pages. If the authors like to keep them, they need to revise them based on my concerns above.

It would be good to show some sample queries and corresponding "meaningful" responses produced by their proposed LocalGAN that are not considered safe responses which are produced by the other tested methods.


DETAILED COMMENTS

For Lemmas 1 and 2 and Theorem 1, the authors need to present them rigorously by specifying the exact math expressions since they have not defined what it means by sufficient, approximates, small enough, and estimate properly. This will also eliminate any discrepancy in their interpretations. For example, the authors have used Taylor series expansion to approximate the expectation of F(q,r) in equation 19 (instead of bounding it). Hence, one can claim that Lemma 2 does not hold and hence Theorem 1 does not hold as well.

In Section 4.3, the described mechanism is confusing to me:

(a) Are the authors saying that it is performed sequentially from foundation to phase-1, followed by phase-2? Or are the authors saying that these three phases are expected behaviors occurring during the optimization in equation 17?

(b) For the foundation phase, is the DBM pre-trained, that is, prior to optimization in equation 17?

(c) Are there multiple response clusters, that is, one for each q? If so, the second RELU term in the minimizing criterion in equation 17 does not seem to properly reflect this.

(d) How are the response cluster centers r_c exactly determined? The authors vaguely say that they are modeled from training data. Is it one center per response cluster? Are the cluster centers also optimized in equation 17, besides the generator's weights? Can the authors provide the argument under the min operator in equation 17? It is confusing to leave out r_c from the subscript of alpha.

I would have preferred that the authors specify the expression of the evaluation metrics to be self-contained.

In Fig. 2, how exactly do the authors measure stability? If the entropy rapidly increases like that of LocalGAN and Adver-REGS, are they considered stable?


Minor issues
Page 1: Despite of?
Page 3: inequation?
Page 3: Equation 4 and 5?
Equation 6: The first summation should just be over q, unless there are multiple sets of R_q per q.
tilde{R}_q is not used in equation 6.
Page 4: a limited samples?
Page 5: defined in 3.2?
Page 5: To be consistent, mathbb should be applied to E.
Page 9: valid this aspect?
Page 9: from the the?

**Experience Assessment:**

I have read many papers in this area.

**Review Assessment: Checking Correctness Of Derivations And Theory:**

I carefully checked the derivations and theory.

**Review Assessment: Checking Correctness Of Experiments:**

I assessed the sensibility of the experiments.

**Review Assessment: Thoroughness In Paper Reading:**

I read the paper thoroughly.

---

> ### Author Response · Authors · 2019-11-11
> **Responses to Reviewer #1**
>
> We are grateful for your valuable comments, and the responses to your concerns are given as follows.
>
> 1.	About the formulations and theoretical proofs: We are working on revising the theoretical proofs of Lemma 1&2 and Theorem 1, which will be given in the Final version of our paper (we will upload it ASAP).
>
> 2.	About the Detailed Comments:
>
> 	a.	In Section 4.3, the described three phases are actually expected behaviors occurring during the optimization in Eq. 17 (Eq. 18 in the final version). We are sorry for the confusion and will add the necessary explanations about it.
>
> 	b.	The DBM is pretrained with the unsupervised strategy before the optimization of Equation 17 (Eq. 18 in the final version).
>
> 	c.	In fact, for each query q, there is only one response cluster (no multiple response clusters). The reason is that, indeed, all the query-response corpora for NRG can be naturally organized into the form {q, R_q | R_q ={r_q_i}}, since the query q can be taken as the key of the item (q, r_q). That is, once a query-response dataset is prepared, R_q can be obtained easily by going through the dataset and reorganize it with a dictionary structure. We feel sorry about the confusion caused by the absent details and will complement them in the revised version.
>
> 	d.	As mentioned in Subsection 3.2, r_c can be greedily obtained with \frac{1}{|R_q|}\sum_i r_i . In the practical training process, we use the well-trained DBM to get the estimation of r_c (labeled with \hat r_c). As mentioned above, each cluster has only one r_c. The center is not optimized in Eq. 17 (Eq. 18 in the final version). The details about \hat r_c is of great necessity and will be supplemented in our Final version. We feel sorry about the confusion.
>
> 3.	The expressions of the evaluation metrics: all the evaluation metrics are widely applied ones in the NRG field, and maybe we can give the expressions in the appendix of the final version.
>
> 4.	About the stability: the training stability, in our opinion, can be reflected by two aspects: First, no obvious fluctuations found in the entropy curve, as discussed in Subsection 5.2. Second, the observed gradient of the curve finally tends to be small. It can be seen that, on the Opensubtitle dataset, the curves of Adver-REGS and LocalGAN show stable obviously. By contrast, on the Sina Weibo dataset, the entropy curves of  Adver-REGS and LocalGAN do increase only at the beginning of the training period, however, we can see that at the end of the training, the convergence trend is also obvious. It is true that, indeed, the Chinese corpus brings the great challenge to the model training, and it is a common view that models on Chinese corpora converge more slowly than those on English corpora.
>
> 5.	Minor issues: the minor issues are addressed. Thanks for the suggestions.

---

> ### Author Response · Authors · 2019-11-14
> **Uploaded  the 3rd version with the Lemmas and Theorem refined.**
>
> We have just uploaded  the 3rd version with the Lemmas and Theorem refined, according to your suggestions.

---

### Official Review · AnonReviewer3 · 2019-10-30
**Official Blind Review #3**

**Rating:** 1

**Review:**

In this paper, the author proposed a model to address the training instability of a GAN model on the NRG task. The authors take advantage of an energy based model to measure the distance between a predicted response and the center of all qualified responses. The training process becomes a hybrid one with the original loss function in the beginning followed by the loss that pulls the response to the center of the response cluster later. In general the paper is well written, with experiments clearly showcased improved training stabilities. However, one major flaw in the experiments in that the authors almost only compared diversity measures such as Dist-1, Dist-2 and Ent4. These measures did not take into consideration the matches between the predicted one and the ground truth. The only relevance measure used in this paper is the  Rel., which the authors defined as the average of embedding distances. Such a measure is by no means an objective measure and can't really demonstrate the effectiveness of the model in terms of generating responses that are close to the ground truth. The authors would need to submit results on one of the widely adopted benchmark metrics (e.g., BLEU, ROGUE) or their equivalents in order to demonstrate the quality of the generated response. And this is the main reason of my rating recommendation.


**Experience Assessment:**

I have published in this field for several years.

**Review Assessment: Checking Correctness Of Derivations And Theory:**

I assessed the sensibility of the derivations and theory.

**Review Assessment: Checking Correctness Of Experiments:**

I carefully checked the experiments.

**Review Assessment: Thoroughness In Paper Reading:**

I read the paper thoroughly.

---

> ### Author Response · Authors · 2019-11-11
> **Responses to Reviewer#3**
>
> We must clarify that, actually, there have been a number of studies drawing the apparent conclusion that the word-overlap based metrics (BLEU, ROUGE) are not suitable for evaluating the relevance of the results given by the open-domain NRG models [1-3]. Moreover, it is the embedding-based similarity that can be taken to replace the metrics like BLEU and ROUGE (or at least as the equivalent of them) in the NRG task [4-9].
>
> Meanwhile, the Rel. metric used in our draft is actually the sum of the three embedding-base similarities (greedy, average, extreme) [2], rather than "... the authors defined as the average of embedding distances ...".  The relevance-oriented metric is explicitly detailed in Subsection 5.1, Page 8 in our paper.
>
> To further conduct thorough comparisons, we have added the human evaluation results in the revised version. The human evaluation is believed to be a meaningful supplement for illustrating the capabilities of NRG models.
>
> References:
>
> [1] Chongyang Tao, et al. "Ruber: An unsupervised method for automatic evaluation of open-domain dialog systems." 2018.
>
> [2] Chia-Wei Lui, et al. "How NOT To Evaluate Your Dialogue System: An Empirical Study of Unsupervised Evaluation Metrics for Dialogue Response Generation."  2016.
>
> [3] Ryan Lowe, et al. "Towards an automatic turing test: Learning to evaluate dialogue responses." 2017.
>
> [4] Yizhe Zhang, et, al. Generating informative and diverse conversational responses via adversarial information maximization. 2018.
>
> [5] Xiaoyu Shen, et, al. "Improving variational encoder-decoders in dialogue generation." 2018.
>
> [6] Iulian Vlad Serban, et, al. "A hierarchical latent variable encoder-decoder model for generating dialogues." 2017.
>
> [7] Yu Wu et, al. "Response generation by context-aware prototype editing."  2019.
>
> [8] Ziming Li, et al. "Dialogue generation: From imitation learning to inverse reinforcement learning." 2019.
>
> [9] Pandey Gaurav, et al. "Exemplar encoder-decoder for neural conversation generation." 2018.

---

### Public Comment · ~Zhengdong_Lu2 · 2019-10-23
**Inspirational paper, and I enjoyed reading it.**

A very inspirational paper, and I enjoyed reading it.

NRG, when casted into a distribution estimation problem, is quite elusive, but it has its own structure that sometimes can be effectively captured to improve the generative performance. The generic GAN,  although aiming to estimate distribution of complicated structure (or weird shape),  does not have the equipment to express our prior knowledge or preference on the distribution.

LocalGAN, however,  attempts to model the distribution in a more explicit way （more specifically through cluster structure）.  The basic assumption of clusters, although quite simple,  partially captures some interesting properties of NRG, like diversity in different dimensions and “interestingness”.   This is in contrast with machine translation , for which the distribution should probably focus on the alignment of words and semantic equivalence of different scales.   That being said,  I still think the authors in their future work can dig deeper to discover more interesting intrinsic structure (eg, linguistic, topic, emotion etc) .

The introducing of adversarial learning into discrete space and the extra mechanism to encourage interestingness indeed makes the learning a bit involved, but the authors seem to solve the problem quite effectively. I didn’t check the equations carefully enough, but I tend to think they did it in the right way.

At last,  I’d recommend the authors to give more cases generated by lcoalGAN,  which, apart from being the most entertaining and enlightening part to  me, also gives clue about which mechanism is picked up by the algorithm.

---

> ### Author Response · Authors · 2019-10-24
> **Reply to “Inspirational paper, and I enjoyed reading it.”**
>
> Thanks for the valuable suggestions.
>
> we will give the cases generated by all the models trained on both datasets, and upload the revised paper with the cases as soon as the rebuttal period begins.

---

### Author Response · Authors · 2019-11-06
**First revision; adding generated cases and revising several issues of equations.**

We have uploaded the 1st version of our revised paper, with the cases generated on both the datasets, according to the suggestions of Reviewer #1 and other readers.
Besides, the following issues of equations are also addressed in this version:
a) all the notations of mathematical expectation in the equations are changed from E to \mathbb E for better consistence;
b) Inequality 6 (Inequality 5 in the Final Version) is carefully revised, including modifying the summation over q, the deletion of \tilde{R}_q, etc., according to the comments of Reviewer #1 and #2;
Meanwhile, some typos are also modified.

It should be noted that, this is NOT the final revised version, and we will continue revising the paper and keep uploading the new versions in the next days.

** As for the Github Link of our code, we need to clarify that all the codes and the link itself have been already anonymized carefully. And the term Kramgasse49 in the link is actually the address of Einstein's House which means nothing to our code. The shared datasets are anonymized as well. **

---

### Author Response · Authors · 2019-11-11
**The second version with Human Evaluation is uploaded**

According to the suggestions of reviewers, we have just uploaded a new version with Human Evaluation results (see subsection 5.3), which can be a meaningful supplement to illustrate the capabilities of models.

Besides, we are now working on revising the formulations and theoretical proofs in our draft, and the Final version will be uploaded soon.

---

### Author Response · Authors · 2019-11-14
**Summary of major revisions.**

In the previous uploaded three versions, we have conducted the following major revisions:

a) Supplemtented the generated cases of all the models trained on both the Opensubtitle and the Sina Weibo Corpora (corresponding to the comments of Reviewer #1).

b) Supplemented the human evaluation results in Subsection 5.3 (corresponding to the comments of Reviewer #2 and #3).

c) Refined the presentation and proof of Lemma 1&2 and Theorem 1 (corresponding to the comments of Reviewer #1).

Besides, the other revisions are detailed in the previous comments from authors.

p.s. : the numbering of equations may be changed due to the modifications.

---

### Decision · Program_Chairs · 2019-12-19

**Decision:**

Reject

**Comment:**

This paper tackles neural response generation with Generative Adversarial Nets (GANs), and to address the training instability problem with GANs, it proposes a local distribution oriented objective. The new objective is combined with the original objective, and used as a hybrid loss for the adversarial training of response generation models, named as LocalGAN. Authors responded with concerns about reviewer 3's comments, and I agree with the authors explanation, so I am disregarding review 3, and am relying on my read through of the latest version of the paper. The other reviewers think the paper has good contributions, however they are not convinced about the clarity of the presentations and made many suggestions (even after the responses from the authors).  I suggest a reject, as the paper should include a clear presentation of the approach and technical formulation (as also suggested by the reviewers).